# Memory-Aware Software Engineering Agents: Architectures, Mechanisms, and Open Gaps

Michael Banf, Johannes Kuhn, Slawomir Garcarz and Max Becker

Perelyn GmbH, Munich, Germany
`michael.banf@perelyn.com`

**Abstract.** AI-assisted software engineering has progressed from code completion to agentic systems operating across the full development lifecycle. Yet these agents remain fundamentally stateless, i.e. each task begins without memory of prior sessions, resolved bugs, or accumulated project understanding. We present a survey that synthesizes evidence from a feature-level analysis of ten production software engineering harnesses with a specific focus on the current state of the art in memory architecture for software engineering agents research. Our analysis reveals an episodic-temporal deficit, i.e. no production harness ships episodic memory or temporal versioning natively. Interestingly, a rapidly growing ecosystem of MCP-based memory add-ons has emerged to fill the deficit. Yet, even these rarely implement temporal reasoning. Similarly, within the research community, while addressing episodic memory only a small fraction consider temporal mechanisms. At the same time, benchmarks confirm a factual-to-cognitive gap across production memory architectures. We argue that persistent, temporally grounded memory is the prerequisite for software engineering agents that learn alongside the teams they serve.

**Keywords:** Software Engineering Agents · Agentic Memory · Context Engineering · Temporal Reasoning · Knowledge Graphs

## 1   Introduction

Agentic AI systems now operate across the full software engineering lifecycle [1,2,3]. Whether resolving complex bugs through multi-file analysis, discovering architectural patterns, or coordinating code reviews across a team, these agents accumulate significant task-specific knowledge during each session. Yet that knowledge vanishes when the session ends: the next invocation begins from scratch, repeating investigations, rediscovering constraints, and re-learning project conventions.

The field has responded rapidly to the challenge [5,6], with every major AI provider now shipping persistent memory. Interestingly, no production harness implements episodic memory or temporal versioning natively, yet a thriving ecosystem of community-built MCP memory add-ons has emerged, evidence that practitioners feel the deficit acutely. However, fundamental properties such as temporal versioning, team-shared memory, or decision provenance are only rarely addressed.

## 2    Memory in Production Software Engineering Harnesses

The CoALA taxonomy [4] maps human memory types onto agent components, i.e. *working*, *semantic*, i.e. conventions, facts, *episodic*, such as past debugging experiences, and *procedural*, such as workflows or tool usage. Thereby, working memory is well-served by modern context windows and semantic memory is partially addressed by context files [7]. However, episodic and procedural memory are almost entirely absent.

A finer-grained functional taxonomy [5] distinguishes *factual*, i.e. what an agent knows, from *experiential*, i.e. what the agent has learned from doing, and *working* memory, i.e. active context management, with all three being critical for software engineering agents operating over long project horizons. From this perspective, we analyzed the memory mechanisms of ten widely-used software engineering agent harnesses. Table 1 summarizes the comparison across 20 dimensions grouped into six categories, i.e. CoALA memory types, memory operations, temporal mechanisms, software engineering specific capabilities, collaboration & governance, and integration.

All ten harnesses support human-written instruction files, representing declarative semantic memory, with widespread adoption [7], yet little resolution improvement [8]. Context engineering [45] formalizes the approach, but all implementations remain manually authored without temporal metadata. Five harnesses generate memories automatically (Claude Code, Codex CLI, Copilot, Windsurf, Gemini CLI), but none distinguishes factual from experiential knowledge. Context compaction triggers vary with respect to window utilization, however implementations are lossy, i.e. technical details, intermediate reasoning steps, and failed-approach information are discarded. However, the quality of compaction determines whether memory helps or hinders [21].

No production system supports temporal versioning, i.e. tracking when knowledge was valid, when it changed, or when it became stale. Neither do they support episodic memory, i.e. structured records of past debugging sessions, resolution trajectories, or decision rationale. Notably, those memory capabilities most critical for software engineering agents that learn over time are those that no tool implements.

### 2.1    The community response: MCP memory add-ons

The gap between what harnesses ship and what practitioners need has spawned a rapidly growing ecosystem of community-built memory extensions, primarily via the Model Context Protocol (MCP) [38]. Our survey identifies over 40 MCP-based memory servers (see selection in table 1). The ecosystem spans every CoALA memory type, however with stark imbalances, i.e. semantic memory is near-universal, episodic recall is emerging, while temporal versioning and procedural memory remain nearly absent. Among the most adopted, *claude-mem* [40] hooks into Claude Code lifecycle events for session capture. *AgentMemory* [39] provides hybrid of BM25 keyword search, vector embeddings and knowledge graph retrieval and *Graphiti/Zep* [12] remains the only reviewed production

| | Memory Types | | | Memory Operations | | | | Temporal | | | Software Engineering | | | | | Collaboration & Governance | | | Integration | |
|---|---|---|---|---|---|---|---|---|---|---|---|---|---|---|---|---|---|---|---|---|
| | Semantic | Episodic | Procedural | Auto-Generated | Compaction | Fact/Experiential | Staleness | Temporal Metadata | Temporal Versioning | Bi-Temporal | Code-Structural | Cross-Session | Provenance | Failed Replay | Git-Aware | Team/Collab. | Access Control | Memory Provenance | MCP Support | Local Deploy |
| **Production Agents** | | | | | | | | | | | | | | | | | | | | |
| Claude Code | ✓ | ✗ | ✗ | ✓ | ✓ | ✗ | ✗ | ✗ | ✗ | ✗ | ✗ | ✗ | ✗ | ✗ | ✗ | ✗ | ✗ | ✗ | ✓ | ✓ |
| Gemini CLI | ✓ | ◆ | ◆ | ✓ | ✓ | ✗ | ✗ | ◆ | ✗ | ✗ | ◆ | ◆ | ✗ | ✗ | ✗ | ✗ | ✗ | ✗ | ✓ | ✓ |
| Codex CLI | ✓ | ✗ | ✗ | ✓ | ✓ | ✗ | ◆ | ◆ | ✗ | ✗ | ✗ | ◆ | ✗ | ✗ | ✗ | ✗ | ✗ | ✗ | ✓ | ✓ |
| Copilot | ✓ | ✗ | ✗ | ✓ | ✓ | ✗ | ◆ | ◆ | ✗ | ✗ | ◆ | ✗ | ✗ | ✗ | ✗ | ✗ | ✗ | ✗ | ✓ | ✗ |
| Cursor | ✓ | ✗ | ✗ | ✗ | ✓ | ✗ | ✗ | ✗ | ✗ | ✗ | ◆ | ✗ | ✗ | ✗ | ✗ | ✗ | ✗ | ✗ | ✓ | ✗ |
| OpenCode | ✓ | ✗ | ✗ | ✗ | ✓ | ✗ | ✗ | ✗ | ✗ | ✗ | ✗ | ✗ | ✗ | ✗ | ✗ | ✗ | ✗ | ✗ | ✓ | ✓ |
| Windsurf | ✓ | ✗ | ✗ | ✓ | ◆ | ✗ | ✗ | ✗ | ✗ | ✗ | ◆ | ✗ | ✗ | ✗ | ✗ | ✗ | ✗ | ✗ | ✓ | ✗ |
| Devin | ◆ | ✗ | ✗ | ◆ | ◆ | ✗ | ✗ | ✗ | ✗ | ✗ | ◆ | ✗ | ✗ | ✗ | ✗ | ◆ | ✗ | ✗ | ✓ | ✗ |
| Aider | ◆ | ✗ | ✗ | ✗ | ✗ | ✗ | ✗ | ✗ | ✗ | ✗ | ◆ | ✗ | ◆ | ✗ | ✓ | ✗ | ✗ | ✗ | ◆ | ✓ |
| OpenHands | ✓ | ✗ | ✗ | ✗ | ✓ | ✗ | ✗ | ✗ | ✗ | ✗ | ✗ | ✗ | ✗ | ✗ | ✗ | ✗ | ✗ | ✗ | ✓ | ✓ |
| **MCP Community Add-ons** | | | | | | | | | | | | | | | | | | | | |
| claude-mem | ✓ | ✓ | ✗ | ✓ | ✓ | ✗ | ✗ | ◆ | ✗ | ✗ | ✗ | ✓ | ✗ | ✗ | ✗ | ✗ | ✗ | ✗ | ✓ | ✓ |
| Mem0 | ✓ | ◆ | ✗ | ✓ | ✓ | ✗ | ✗ | ◆ | ✗ | ✗ | ✗ | ✗ | ✗ | ✗ | ✗ | ◆ | ✗ | ✗ | ✓ | ✓ |
| Graphiti / Zep | ✓ | ✓ | ✗ | ✓ | ✓ | ✗ | ✓ | ✓ | ✓ | ✓ | ✗ | ✓ | ✓ | ◆ | ✗ | ✗ | ✗ | ✓ | ✓ | ✓ |
| Cognee | ✓ | ✗ | ◆ | ✓ | ✓ | ✗ | ✗ | ◆ | ✗ | ✗ | ✓ | ✓ | ✗ | ✗ | ✗ | ✗ | ✗ | ◆ | ✓ | ✓ |
| AgentMemory | ✓ | ✓ | ◆ | ✓ | ✓ | ◆ | ✗ | ◆ | ✗ | ✗ | ✗ | ✓ | ◆ | ✗ | ✗ | ✗ | ✓ | ◆ | ✓ | ✓ |
| Letta / MemGPT | ✓ | ◆ | ◆ | ✗ | ◆ | ◆ | ◆ | ◆ | ✗ | ✗ | ◆ | ✓ | ◆ | ◆ | ✗ | ✗ | ✗ | ◆ | ✓ | ✓ |
| Codebase-Memory | ✓ | ✗ | ✗ | ✓ | ✓ | ✗ | ✗ | ✗ | ✗ | ✗ | ✓ | ✓ | ✗ | ✗ | ✗ | ✗ | ✗ | ✗ | ✓ | ✓ |
| Engram-MCP | ✗ | ✓ | ✓ | ✓ | ✓ | ✓ | ✗ | ✓ | ◆ | ✗ | ✗ | ✓ | ✓ | ✓ | ✓ | ✗ | ✗ | ✓ | ✓ | ✓ |
| Ourmem | ✓ | ✗ | ✗ | ✓ | ◆ | ✗ | ✗ | ◆ | ✗ | ✗ | ✗ | ✓ | ✗ | ✗ | ✗ | ✓ | ✓ | ◆ | ✓ | ◆ |
| Hindsight | ✗ | ✓ | ✗ | ✓ | ◆ | ✗ | ◆ | ◆ | ✗ | ✗ | ✗ | ✓ | ◆ | ✓ | ✗ | ✗ | ✗ | ◆ | ✓ | ✓ |

✓ Supported   ◆ Partial   ✗ Absent

**Table 1.** Memory capability matrix across 10 production SE agent harnesses and 10 representative MCP community add-ons, evaluated along 20 dimensions grouped into six categories: CoALA memory types, memory operations, temporal mechanisms, SE-specific capabilities, collaboration & governance, and integration. Temporal versioning, bi-temporal validity, and governance remain near-universally absent.

system with true bi-temporal validity intervals. Software engineering specific servers include *Codebase-Memory* [36], a Tree-Sitter knowledge graph across a plethora of languages, *Engram-MCP* [41], for git-branch-aware session handoffs, or *Hindsight* [42], featuring failed attempt replay. Notably, nine of ten major coding agents now support MCP, making these add-ons agent-agnostic. Built-in memory sophistication varies widely. For example, Gemini CLI offers a four-tier hierarchy with automatic skills extraction, i.e. episodic-to-procedural conversion. Copilot performs just-in-time codebase verification of stored facts, while Cursor has removed its auto-memory feature, suggesting that naive auto-capture without proper architecture degrades quality.

## 3    Memory Architectures: From Extraction to Graphs

Our classification of the ICLR 2026 MemAgents workshop paper collection [6] reveals that retrieval-augmented memory is the dominant paradigm, underpinning over 60% of contributions. Within this paradigm, approaches differ primarily in how they organize, consolidate, and temporally ground stored memories. A second, smaller cluster of non-RAG approaches operates at the model or inference level, including KV-cache compression, attention-based, and embedding-based methods. Cross-cutting learning strategies such as experience replay, reinforcement learning-based management, and hierarchical consolidation span both clusters. Three retrieval-augmented variants have reached production maturity and been formally benchmarked head-to-head on the LoCoMo-Plus cognitive memory evaluation [9], i.e. extraction-based, self-managing, and graph-based temporal.

### 3.1    Extraction-Based Memory

Extraction-based architectures, such as Mem0 [10], extract atomic facts from conversations and store them as vector-indexed entries for similarity based querying. Extraction-based systems excel at *factual recall*. For software engineering tasks requiring explicit knowledge retrieval, this paradigm is well-suited. However, extraction is lossy by design, e.g. the process of converting debugging sessions into atomic facts discards causal chains, temporal context, and reasoning paths [13].

### 3.2    Self-Managing Memory

Self-Managing Memory, such as Letta [11], treats the LLM itself as an operating system managing its own memory via explicit read/write/search operations over a hierarchical store. The agent decides what to remember and how to organize it. Self-managing memory excels at preserving *behavioral continuity*, i.e. reasoning on goals, recurring patterns, and evolving state. For software engineering agents that must maintain an evolving picture of project architecture, track long-running refactoring goals, or remember what patterns emerged during debugging, this is the strongest fit, although less transparent than explicit stores: the agent's internal state cannot easily be inspected, audited, or corrected by developers.

### 3.3    Graph-Based Temporal Memory

Graph-Based temporal approaches, such as Graphiti [12], can distinguish *event time*, i.e. when something happened, from *ingestion time*, i.e. when the system learned it, with every edge carrying explicit validity intervals. This *bi-temporal* approach is unique among production memory systems. Graph-based memory excels at *causal reasoning*. For software engineering tasks requiring understanding of why a decision was made, how events connect across time, or what caused a regression, temporal graphs provide the most natural representation. However, graph construction adds latency and complexity.

### 3.4   Hybrid and Emerging Approaches

The LoCoMo-Plus results also make a compelling case for hybrid architectures, since no single paradigm masters all competency types [9], and it confirms that the systems capture genuinely different aspects of the memory problem. To this end several emerging directions point toward future software engineering relevant architectures. Reinforcement learning based memory management, such as Memory-R1 [26] and AgeMem [27] replace heuristic memory policies with trained ones. The learned policies discover non-obvious strategies such as pre-emptive summarization before the context window fills, suggesting a path toward adaptive memory management tuned to software engineering specific reward signals such as issue resolution rate, code quality metrics. Multi-graph memory, as in MAGMA [23], represents each memory item across orthogonal semantic, temporal, causal, and entity graphs, with retrieval formulated as policy-guided traversal. This decomposition offers fine-grained control particularly relevant for software engineering: dependency relationships via an entity graph, decision provenance via a causal graph, API validity via a temporal graph, and conceptual similarity via a semantic graph can be queried independently. A formal theory of hierarchical memory [46] defines three operators, i.e. extraction, coarsening, and traversal. This identifies a self-sufficiency spectrum for representative functions that constrains viable retrieval strategies. TiMem [24] implements this via a temporal memory tree structure. For software engineering agents processing long execution traces, hierarchical consolidation could be essential. Hybrid local/cloud architectures are also emerging: LightMem [43] delegates memory operations, such as routing or filtering, to small local models while reserving large models for consolidation, and HyMem [44] achieves comparable performance at reduced cost via dynamic two-tier retrieval. Since the majority of MCP memory servers support fully local operation and frameworks like Mem0, Letta, and Cognee run on Ollama, sovereign deployments are feasible, though reliable MCP tool calling requires 14B+ parameter models [35,17].

## 4   Memory Mechanisms for Software Engineering

Beyond general-purpose memory architectures, several mechanisms have been developed specifically for or with direct relevance to software engineering agents.

### 4.1   Versioned Context Management and Decision Provenance

The Git Context Controller (GCC) [14] introduces Git-inspired operations for agent memory, addressing the *intra-session* memory problem, i.e. managing working memory within a single task execution. The Lore protocol [15] addresses the complementary *inter-session* problem as it repurposes git commit messages as structured decision records carrying constraints, rejected alternatives, agent directives, and verification metadata via native git trailers, introducing queryable decision provenance. The practical need for decision provenance is acute. For

example, each additional deviation in a mutating action reduces success odds [30], with errors growing as context length increases and agents drift from stale constraints, such as deprecated APIs [28,29].

## 4.2   Experiential and Cross-Session Learning

The SWE Context Bench [21] evaluates whether agents benefit from prior experience with one critical finding, i.e. summarized experience improves resolution, while unfiltered experience hurts. Experiential Co-Learning [16] mines experiences from historical trajectories and injects them into future task execution. Experiential Reflective Learning [17] reflects on task trajectories to generate transferable heuristics, i.e. actionable lessons that generalize across tasks. Live-SWE-agent [18], rather than remembering past experiences, self-creates new tools during execution, representing procedural memory through self-modification. Structurally Aligned Subtask-Level Memory [19] decomposes software engineering problem-solving into subtasks and aligns memory storage and retrieval at the subtask rather than the whole-instance level. MemGrad [20] transforms batches of behavioral feedback into a dual memory strategy, i.e. retrospective memory captures recurring failure modes, while prospective memory encodes gradient-derived strategies for future reasoning. Procedural memory, such as $Mem^p$ [37], distills past agent trajectories into both fine-grained step-by-step instructions and higher-level script-like abstractions, creating learnable, updatable procedural memory. Procedural knowledge built from a stronger model transfers effectively to weaker ones. Strategy-level memory, such as ReasoningBank [47], rather than storing raw trajectories, distills generalizable reasoning strategies from self-judged successes and failures. Finally, persistent note-taking [48] introduces a cross-session note-taking system where agents record hindsight notes for compilation errors, runtime exceptions, and unproductive strategies. WebCoach [35] reveals a striking capacity threshold for experiential memory: 7B models do not benefit from experiential memory, while 32B+ models show pronounced gains. Self-generated experiences outperform externally seeded ones [17]. For software engineering agents, this implies that experiential memory requires both sufficient model capacity to reason over retrieved experience and careful curation to avoid context pollution.

## 4.3   Knowledge Graphs for Software Engineering Agent Memory

A comprehensive survey on graph-based agent memory [22] identifies four lifecycle phases, from extraction, that is converting raw data into graph nodes and edges, followed by storage, i.e. organization and indexing, then retrieval, i.e. query-adaptive traversal, and evolution, i.e. update, consolidate, forget. For software engineering agents, graph memory offers three specific advantages. First, they naturally implement *dependency tracking* via relationships for inherently graph-structured code repositories, including files import modules, classes inherit from others, functions call functions [36]. Second, they provide *decision provenance*. Architectural decision records, code review discussions, and commit rationale form a causal graph of project decisions. Graph-based memory can model and

preserve the reasoning behind code structure, enabling agents to understand constraints before proposing changes. Third, they lay a foundation for monitoring *temporal evolution*. For example, Graphiti's bi-temporal model maps naturally onto software evolution, such as an invalid API, a recently adopted coding convention or deprecated dependency. GAM [25] demonstrates the value of hierarchical graph memory showing significant improvements over extraction-based approaches by decoupling episodic from semantic consolidation via an event progression graph.

### 4.4   Collaborative and Team Memory

Software engineering is inherently collaborative, yet memory research exhibits a strong single-agent bias [6]. To this end, Collaborative Memory [31] is the most complete framework for multi-user memory sharing with dynamic access control. It maintains two tiers. These are *Private memory*, which is visible only to the originating user and *shared memory*, which consists of selectively shared fragments, where each fragment carries immutable provenance attributes, i.e. contributing agents, accessed resources, timestamps. This architecture maps naturally onto software engineering teams w.r.t private developer context, i.e. local experiments, debugging notes, versus shared project knowledge, i.e. coding conventions, architectural decisions, resolved issues. However, MINJA [32] demonstrates that shared memory agents are vulnerable to query-only injection attacks. For software engineering agents with access to codebases and deployment pipelines, memory poisoning is a critical supply-chain risk. SSGM [33] and MemArchitect [34] propose governance middleware for evolving memory, while the mnemonic sovereignty survey [49] addresses EU AI Act compliance. Yet no approach targets software engineering-specific governance needs, such as access control aligned with repository permissions, memory provenance linked to code review approvals, or temporal policies tied to release cycles.

## 5   Discussion

Synthesizing our analysis, we identified several open gaps specific to memory for software engineering agents as inspirations for future research. Software engineering agents discard resolution trajectories after each session. Early research solutions exist, but no production harness or community add-on integrates cross-session learning. No production software engineering agent and only a tiny fraction of community add-ons track temporal knowledge validity. No software engineering specific temporal memory exists. Also, there exists only one software engineering specific temporal benchmark. Convention retention, experience transfer, staleness detection, and decision provenance remain unevaluated. Collaborative memory is rarely addressed, yet software engineering teams need shared decisions, private debugging notes, and repository-aligned access control. Research on memory injection attacks demonstrates that shared-memory agents lack adequate governance and security, while GDPR-compliant deletion remains entirely unaddressed.

Software engineering craft knowledge is overwhelmingly procedural, yet procedural memory is the rarest type in both research and the community ecosystem. The MCP ecosystem provides a composable memory architecture where each server addresses a different CoALA memory type. Yet no orchestration layer decides which memory to consult when, resolves conflicts between overlapping stores, or manages lifecycle across layers. Assembling the full stack remains a manual integration challenge. Closing these gaps will require a coordinated effort and a clear vision of transforming software engineering agents from amnesic tools into learning collaborators that grow alongside the software they help build.

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
