# OpenReview forum: "Memory-Aware Software Engineering Agents: Architectures, Mechanisms, and Open Gaps"
_KI/2026/Workshop/AI4SE — AI4SE Workshop_

### Official Review · Reviewer_pkor · 2026-06-08
**See below.**

**Rating:** 6
**Confidence:** 4

**Review:**

This paper surveys memory architectures for software engineering agents and argues that, despite growing production support and MCP-based add-ons, current systems still lack persistent episodic and temporally grounded memory needed for agents to accumulate project understanding across sessions.

Points in favor:
- Highly relevant topic.
- Good overview of found approaches.

Points against:
- Research goal was not stated clearly.
- Methodology of how survey was performed is not clear.
- Paper is lacking discussion of limitations of work performed.

Further remarks regarding sections:

Abstract
- Episodic memory and temporal versioning should be explained a bit as central concepts.

Introduction
- The end of the first paragraph seems to be too general: Claude code documents auto memory across sessions. Copilot uses repository or custom instructions as reusable project context.
- Some terms, such as MCP, episodic memory and temporal versioning should be explained more in the introduction (Section 2 explains memory types in more detail).
- Goal of research should be mentioned in introduction.

Memory in Production Software Engineering Harnesses
- Section is a combination of background knowledge and study results.
- Section is missing a description of the method performed to collect all that information shown in Table 1. This makes it quite untransparent how the results were obtained: How was the survey performed? What was included/excluded based on what criteria? How was the rating / evaluation of the different approaches done?

Memory Architectures: From Extraction to Graphs
- How are the columns in Table 1 related to the memory architectures presented? Obviously, some terms overlap and Table 1 (as a major result) should be referred to in this section as well.

Memory Mechanisms for Software Engineering
- Same comment regarding Table 1 as for previous section.

Discussion
- Research gap is described based on previous analysis. However, there is no discussion of any limitations or threats to validity of the study performed.

---

### Official Review · Reviewer_jtpy · 2026-06-09
**The paper provides an interesting contribution and serves as a valuable basis for discussion for the proposed workshop.**

**Rating:** 7
**Confidence:** 4

**Review:**

The paper addresses the statelessness, or lack of memory, of agents used in AI-assisted software development. It highlights the challenge that knowledge and insights generated during development are typically not retained, preventing the accumulation of temporally referenced experience throughout the software lifecycle. In addition to analyzing specific software engineering agents, the authors explore data structures that could be used for memory architectures and discuss their potential application within software engineering contexts.
Potential suggestions for improvement:
- The analysis of software engineering agents and MCP extensions presented in Chapter 2 constitutes the empirical core of the paper. The six evaluation factors and the derived assessment criteria should be explained and justified in greater detail.
- The paper relies heavily on textual descriptions, which makes the content of Chapters 3 and 4 somewhat difficult to grasp. For the final version, the inclusion of explanatory figures and summary tables would improve readability and comprehension. Additionally, the standalone Section 2.1 should be integrated into the surrounding structure.
- While the topics covered are highly relevant and interesting, the paper could benefit from a stronger focus and a more clearly defined scope.